# Unveiling the Essential Role of Green Spaces during the COVID-19 Pandemic and Beyond

Mariusz Ciesielski [1,*], Piotr Gołos [1], Fruzsina Stefan [2] and Karolina Taczanowska [2]

1. Forest Research Institute, 05-090 Sękocin Stary, Poland; p.golos@ibles.waw.pl
2. Institute of Landscape Development, Recreation and Conservation Planning, University of Natural Ressources and Life Sciences Vienna, Peter-Jordan-Strasse 82, 1190 Vienna, Austria; fruzsina.stefan@boku.ac.at (F.S.); karolina.taczanowska@boku.ac.at (K.T.)
* Correspondence: m.ciesielski@ibles.waw.pl; Tel.: +48-227150328

**Abstract:** The COVID-19 pandemic highlighted the essential role of urban and rural green spaces for societies coping with global public health crisis. During this particular time, a significant body of research was devoted to human–nature relationships, as well as the use and importance of green spaces, both from the management and visitors' perspectives, along with the vital role of nature in human health and wellbeing. Furthermore, the pandemic experience induced new paradigms in spatial and urban planning, along with the management of forest and protected areas seeing the crucial role of green spaces in shaping long-term socio-environmental resilience and sustainability. Thus, after the official end of the pandemic, our study aimed to provide a systematic review of the international research related to green spaces within the context of the COVID-19 pandemic, focusing on those published between 2020 and 2023. The literature search within SCOPUS and Web of Science databases was conducted on 16 May 2023. A dataset of 161 articles was analyzed using a two-stage analysis. In the first stage, screening based on the title, abstract, and keywords was carried out. In the second stage, a detailed full text analysis was carried out, resulting in a final dataset of 66 articles related to the scope of this review. This article gives an in-depth methodological and conceptual overview, also referring to the applied research and management context related to green spaces in urban and rural environments. It concludes with lessons learned and poses open questions for future research related to green space planning and management. The literature review shows that institutions managing green spaces in cities and forests are facing new challenges. These include pursuing sustainable management policies in cities, ensuring equitable access to urban green space and community participation in the decision-making process, adapting suburban forest management to social expectations, and the recreational development of forest areas taking into account social needs and ecosystem sustainability.

**Keywords:** forests; urban green spaces; recreation; COVID-19; human mobility; well being

## 1. Introduction

Forests are an integral part of landscapes and have fulfilled numerous functions for centuries. Originally, the production function dominated, but since the beginning of the 20th century, the non-productive functions of forests (social and protective) have also been increasingly recognized [1]. Urban green spaces, which include all managed green spaces (parks, squares, green areas) and unmanaged green spaces (urban forests, woodlands, wastelands), are also important for society and an important element of the urban composition [2,3]. The role of forests and green spaces in society is reflected in the concept of ecosystem services. According to the international classification of ecosystem services, forest areas are considered capable of providing more than 100 ecosystem services [4]. Ecosystem services are divided into four groups: cultural, regulating, provisioning, and supporting [5]. The need for services and the demand for them are diverse [6]. Hegetchweiler et al. [7] showed that nonmaterial ecosystem services are more important to society,

especially for residents of urban areas, than those that provide specific products. Among the cultural services, those associated with the possibility of active and passive recreation in green spaces and contact with nature are of particular importance [8]. As research published over several decades has shown, contact with nature has a positive effect on human health and well-being [9,10]. Human health is also influenced by regulating ecosystem services that reduce the risk of lung diseases through air purification, among other things [7]. However, in urbanized areas, access to green spaces is usually limited or unevenly distributed across urban areas. Nevertheless, publicly accessible green spaces and forests are important public spaces that serve as places for daily recreation [11]. The importance of these places was demonstrated in the early days of the COVID-19 pandemic.

The COVID-19 pandemic officially ended, after more than 3 years, on 5 May 2023, according to the decision of the World Health Organization [12]. This period was characterized by several waves of disease and series of decisions taken by individual countries to prevent the disease [13]. The restrictions imposed by governments affected practically all aspects of life [14,15], as they consisted of limiting contact and ordering social distancing, closing state borders, banning movement, and closing schools, kindergartens, and workplaces; remote teaching and remote work were introduced on a scale unprecedented up to the onset of the pandemic. The impact of these restrictions was reflected in changes in society's behavior [16–18]. Among other things, the rhythm of daily life has changed [19], and many people have felt worse or fallen into depression [20]. Restrictions on carrying out usual life activities indoors and the sudden growth in free time have led to an increased interest in contact with nature [21–23]. Society has found that green spaces have become one of the few places where people can spend their free time, passively or actively. Even in the first weeks of the pandemic, there was a significant increase in the number of visitors to green spaces [21]. In some countries, such as Austria and Poland, this increase induced access limitations to some urban green spaces and forests [24]. Several social groups have rediscovered the benefits of outdoor recreation or decided to make such visits for the first time during the pandemic [23].

Research related to COVID-19 pandemic has highlighted the crucial role of green spaces for society. Studies conducted in various countries referred to the recreational use of green spaces, including the aspect of human mobility, frequency and duration of visits, and choice of recreational sites [25,26]. Researchers also focused on studies that described the effects of green spaces on people's health and well-being during times of public health crises [27,28]. Another group consisted of studies on the spatial policies of cities and regions with respect to the availability of green spaces for recreation [29,30]. The papers also presented numerous challenges for agencies responsible for green spaces management in terms of adapting their activities to new social expectations (e.g., increasing the role of nature-based solutions and green infrastructure, designing green spaces taking into account the needs of different user groups, ensuring equal access to green spaces) [31,32].

The end of the pandemic period is a great opportunity to sum up three years of research focused on different aspects related to the role of green spaces during the COVID-19 pandemic. The objective of this study was to conduct a systematic review of the literature to attempt to answer the following questions:

- What research methods have been used to determine the social importance and ways of use of green spaces?
- Did the recreational use of green spaces in cities and rural areas change during the pandemic, and how?
- What impact did visiting green spaces have on people's health and well-being?
- What impact may the pandemic have on land-use policies and decisions made by managers of green spaces and forests?

## 2. Materials and Methods

A systematic literature search was performed in the SCOPUS and Web of Science (WoS) databases (Figure 1). Articles, books, and conference proceedings on this topic were searched according to the methodology described by Pullin and Stewart [33], Snyder [34], and Grant and Booth [35]. The same four search queries were entered into both databases using the following keywords: (1) "forest", "recreation", "COVID-19"; (2) "green spaces", "recreation", "COVID-19"; (3) "green areas", "recreation", "COVID-19"; (4) "ecosystem services", "recreation", "COVID-19". Keywords were linked using the logical conjunction "AND" and searched in the subsequent fields: "title", "abstract", "keywords". Papers in all languages were searched in both databases. Due to the limited time frame of the research topic, the period in which the literature should be searched was not determined. The search was conducted on 16 May 2023.

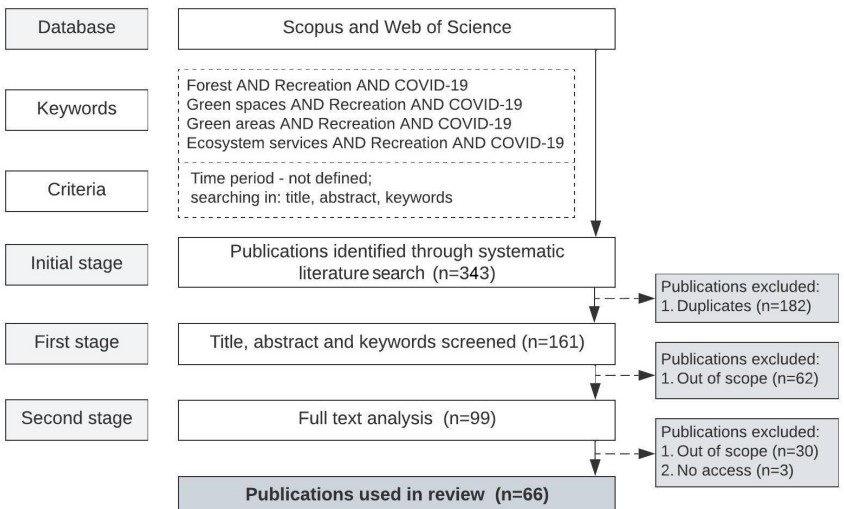

**Figure 1.** The procedure of searching for articles, books, and conference materials that meet the adopted analysis criteria.

For the selected literature articles, a full bibliometric description was downloaded from the databases in a .csv file. In total, all queries in the SCOPUS database returned 163 articles and 180 in the Web of Science database. After combining the obtained databases, duplicate records were removed, leaving 161 articles for a further detailed two-stage analysis. In the first stage, based on the title, abstract, and keywords, articles were selected for further analysis, the topics of which were as follows:

- Issues of social mobility in forest areas and green spaces during the pandemic COVID-19;
- Recreation in forests and green spaces during the pandemic COVID-19;
- Health issues related to presence in green spaces;
- Implications of COVID-19 for strategic planning forest or spatial planning policies for urban areas.

Articles written in a language other than English or Polish were excluded. There were a total of 62 articles that did not cover the relevant topics. A total of 99 articles were qualified for the second stage of analysis.

In the second stage, after analyzing the content of the articles based on thematic criteria, 30 articles were excluded because of thematic incompatibility. In addition, three articles that were not published via the open access formula were not included in the review. Finally, 66 papers were analyzed in detail and included in the literature review.

## 3. Results

### 3.1. Bibliographic Overview and Study Methods

The subject has been a topic of research in 31 countries on five out of seven continents. No publications were found for South America and Antarctica (Figure 2). Most of the papers were published in the United States (10 papers), followed by Italy and Poland (6 papers each), and Germany and Norway (4 papers).

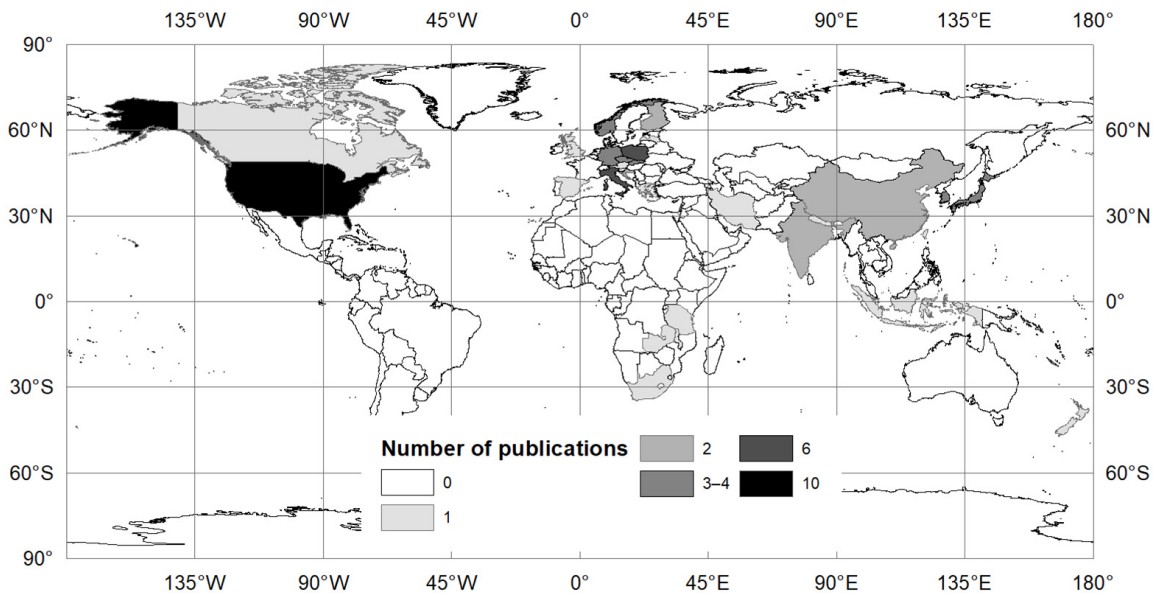

**Figure 2.** Geographical distribution of published items on this subject.

In the articles analyzed, the most common keyword was COVID (41 occurrences) (Figure 3), followed by forest(s) (38), recreation (26), and urban (26). Words used also included health, well-being, and availability of green area.

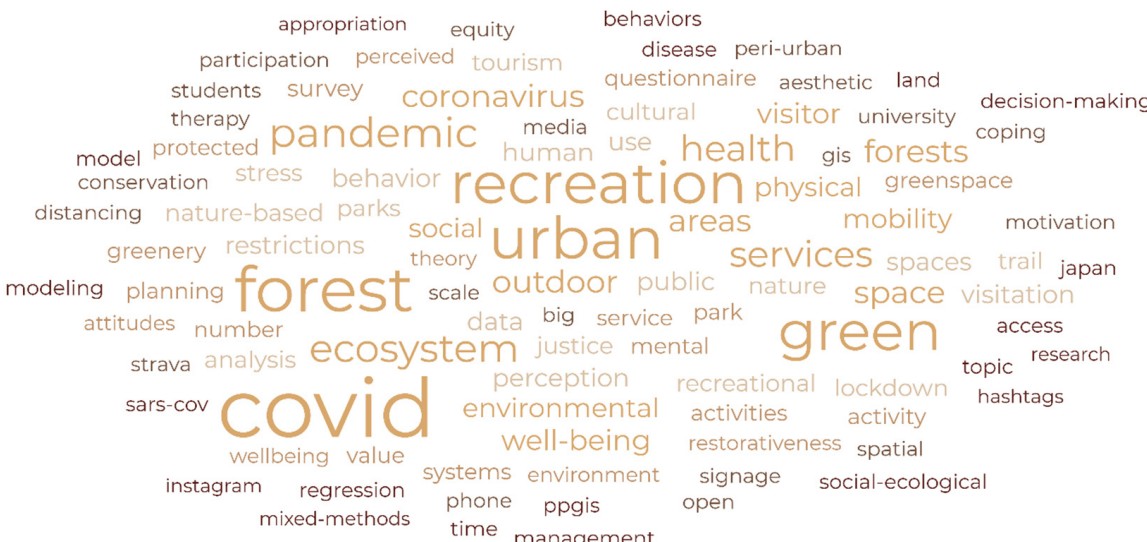

**Figure 3.** The frequency of keyword occurrences.

The study comprised various types of green spaces. Based on the dominant green area in the article, a subdivision was made into the following categories: forests (forests outside urban areas and in their immediate surroundings or forests without a specified spatial reference, which include all forest areas, e.g., in the country) (13 papers) [36],

suburban and urban forests (forest areas within cities or in their immediate surroundings) (11 papers) [37], various areas (the authors did not specify a predominant type of green space but analyzed many different ones, e.g., it could be forests, parks, national parks, wasteland) (12 papers) [38], and urban green spaces (UGS) (the authors clearly defined that they were concerned with UGS, without naming the predominant type of green that was the subject of the research) (30 papers) [39]. Society's perception of the pandemic changed during its duration (from fear to denial or indifference) [40]. Therefore, studies have been carried out on the importance of green spaces for society during the different phases of the pandemic and the restrictions introduced (lockdown during the first wave of the epidemic, lockdown during later waves, phase of easing restrictions, etc.) [21,24,37,38]. Researchers have emphasized that restrictions imposed by governments, including those that minimize disease risk, have influenced the choice of research methodology. The vast majority of studies were conducted remotely or with appropriate social distance. The most commonly used method was survey research (35 papers) [41], and two studies additionally combined survey research with a built-in map module (called participatory geographic information system (PPGIS)) [42]. Eleven studies used a combination of two or more methods, called data fusion [43–45]. Data generated by application and social media users were used four times [46], while mobile data were used three times [14]. In addition, researchers used field observations, camera traps, and measurement sensors (three papers each) [47,48], information on the number of overnight stays (one paper) (other category) [49], and data from electronic payment systems (one paper) (other category) [50] (Figure 4).

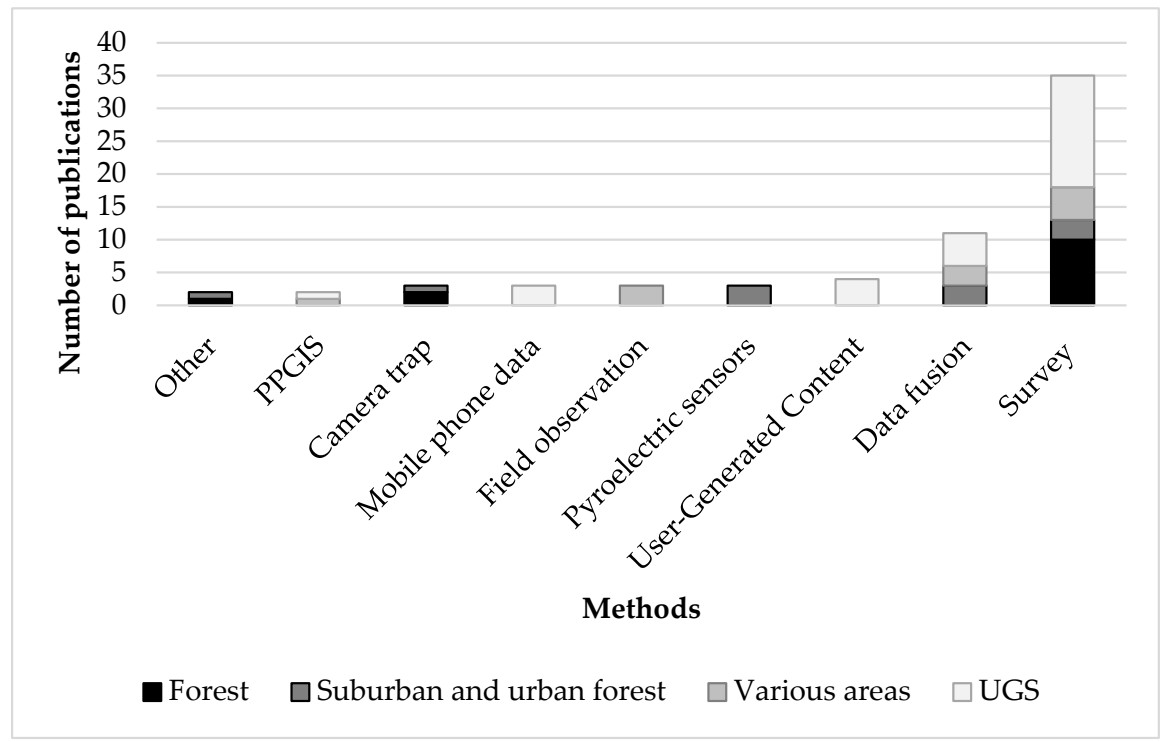

**Figure 4.** Distribution of analyzed publications by research area and research method.

Below, the basic characteristics of the methods used are described.

- Surveys research

Surveys are one of the most frequently used methods to obtain data on society's stated preferences, including those regarding the recreational use of natural areas [51]. This method was also most commonly used during the pandemic (35 papers), especially in online forms due to the applicable restrictions. The number of respondents varied from 47 [52] to 3286 [53] (an average of 807 people). The surveys were conducted in areas at different

scales: local (e.g., a single city) [23], regional (e.g., a metropolitan area) [33], national (e.g., studies covering the whole country) [43], or continental (e.g., studies conducted in different countries) [22]. Due to the online form of the surveys, it was very difficult for the researchers to obtain a representative sample of respondents [54]. The surveys were conducted at different times during the pandemic and the restrictions introduced, which made it possible to observe changes during the pandemic [23,55]. Most of the work for which surveys were carried out (29.4%) concerned the spring of 2020, when the first wave of the pandemic occurred and when it was decided to introduce restrictions for the first time. In total, 26.5% of the surveys were conducted during the period of easing restrictions (summer 2020, 2021) and during the second or third wave of the coronavirus in spring 2021. The surveys not only provided information on the reported frequency of visits to natural areas, but also were the source of data that allowed us to learn about the factors that motivate people to spend time in natural areas [22], the demand for green spaces [53], the effects of green spaces on the psychophysical state of society [56], or the effects of restrictions on recreation [24].

- Public Participation GIS (PPGIS)

Survey research using the map module made it possible to obtain detailed information about the place of residence and the places important to respondents or places visited for recreational purposes. This method was used in a study by Korpilo et al. [42] and Fagerholm et al. [57] (two papers). In the first study, the authors examined a sample of respondents who were surveyed for the first time in 2018. This made it possible to compare results from before the pandemic and during the pandemic. Fagerholm et al. [57] designated green spaces that differed in the level of social activity during the pandemic.

- User-generated content and mobile phone/mobile device data

Data generated by mobile phone users can be divided into two groups. The first group includes data created by users of social media and applications (four papers). Some of them are publicly available and contain information about the date and place of their execution (description or geographical coordinates). Research on leisure activities during the pandemic used quantitative and qualitative data (analysis of descriptions, content, photos) from Twitter [31,58], Instagram [46], or Strava [45]. The second group consists of data collected directly by mobile operators based on the signals sent by phones and recorded by reference stations (three papers). This group also includes data provided by commercial companies based on anonymously collected data from mobile devices. Examples of research using mobile data during a pandemic include the work of Kim et al. [25] and Day [38]. These data provided information on the number of visits and their durations.

- Direct observations in the field

Field observations at designated sites, along trails and paths, were conducted by trained staff (three papers). Because of the restrictions introduced during the COVID-19 pandemic, it was critical to properly position observers while maintaining sanitary rules. This method was used for observations primarily aimed at verifying that visitors staying in natural areas were following recommendations such as wearing masks or maintaining social distance [48,59,60].

- Indirect observations—pyroelectric sensors

Pyroelectric sensors are one of the methods of continuous monitoring. They provide quantitative, point-by-point information on the number of visits to natural areas. Their appropriate calibration with ground observations and their distribution in the study area allow for modeling the mobility of society. During the pandemic (three papers), this method was used to compare the number of visits during the entire pandemic period, divided into phases that differ in the extent of restrictions, with the period or years before the pandemic [37,47]. However, both studies used a small number of sensors, meaning that the results represent local changes on selected trails.

- Indirect observations—camera traps

Data collected with camera traps belong to the methods of continuous monitoring of punctual character. They provide mainly quantitative information, but it is also possible to obtain qualitative information. This is possible with manual or automatic methods of image content analysis. With this method, it is important to place the measuring instruments correctly and ensure compliance with GDPR regulations. This method is mainly used at the local level. In this review, the use of camera traps (three papers) is shown in the studies conducted by Dudáková et al. [61], who collected data on visitation and use of forest roads. They provided quantitative information on recreational use during the pandemic on daily, monthly, and annual bases. Based on the analysis of the collected material, qualitative data were also obtained, ensuring the distinguishing of modes of transportation (e.g., bicycles, all-terrain vehicles, passenger cars, private vehicles, and Forest Service vehicles). Cukor et al. [62] and Procko et al. [63] used data from camera traps to examine the effects of changes in the number of forest visitors on the behavior of selected wildlife species.

- Data fusion

Based on the advantages and limitations of the methods mentioned in subsections 1–6, it was also decided to use various combined methods in the research (11 papers). For example, quantitative data were compared [44], user-generated content data were corrected with regression models based on data from ecounters [45], and quantitative data were collected using ecounters. These data were then combined with qualitative data, e.g., on declared motives influencing the decision to visit forests [43].

In summary, well-known research methods were used during the COVID-19 pandemic. A great emphasis was placed on methods and instruments that allowed for remote data collection. The methods presented were used at different scales, from local (e.g., trails, park, city) [64] to regional [49], national (e.g., forests in Poland, the Czech Republic) [65], or continental (e.g., surveys in many countries) [22] scales. Due to the diversity of studies conducted, it is difficult to specify a universal method that should be used in situations with limited contacts. However, Venter et al. [66] emphasize that currently, data with high a spatial and temporal resolution enable the appropriate mapping of both the demand for ecosystem services (recreation) and the possibility of providing them through green spaces (supply). It is extremely important to conclusively present the benefits of particular areas. However, it should be noted that it is important not only to look at the number of visitors to green spaces quantitatively, but also to obtain qualitative information that allows for the needs of the different user groups to be accurately identified.

*3.2. The Importance of Green Spaces and Forests as Relatively Safe Places for Recreation during the COVID-19 Pandemic*

Society's perception of the pandemic changed over time (from fear to denial or indifference) [29]. These reactions, as well as the gradation of the restrictions introduced, had a significant impact on the perception of the role and importance of green spaces and forests. Most of the publications analyzed refer to 2020, when restrictions very often took the form of lockdowns [21]. In addition, the results of numerous studies conducted in different places and at different times indicate an increase in the importance of green spaces as places for recreation and relaxation—these spaces are perceived as places that have a positive impact on health. Weibrenner et al. [23] even suggested in the title of their paper that "The forest has become our new living room". Green spaces became a kind of substitute resource during the lockdown as other places of recreation were closed [67]. The significance of green spaces during the first phase of the pandemic in spring 2020 can also be seen in the negative reaction of society in Poland to the introduction of a short-term ban on entering green spaces and forest areas. Baranowska et al. [68] indicated that more than 70% of respondents were critical of such a restriction. Similar results were presented by Kikulski [24], who also attributed the negative social reception of this restriction to the time of year—spring is the second most popular season for forest recreation in Poland [46].

Criticism of the introduced restrictions was probably also a consequence of the lockdown of recreational and leisure facilities such as cinemas, gyms, or playgrounds.

In the subsequent phases of the pandemic, the role of green spaces was also important for society. Day [38] emphasized in his research that despite the reduction of restrictions and the return to normal life, interest in visiting green spaces was still greater than that in the years before the pandemic. This could prove that the lockdown was a kind of catalyst for the increased interest in recreation in green spaces. De Mao et al. [69] conducted their study during the third wave of the infection, a time when respondents had a better appreciation of the importance and role of suburban forests based on their own experiences from two previous lockdowns. According to the authors, the period in which the survey was conducted largely eliminated the influence of emotions on the answers given. It should, therefore, be recognized that the opinions expressed by respondents in this study about the role of green spaces as a place for recreation during the pandemic are significant.

3.2.1. Mobility of Society and Demand for Recreation in Nature

Surveys in Indian cities have shown that UGS use is increasing. The percentage of respondents who declared daily visits to UGS areas increased by 9% after the lockdown (Table 1). There was also an approximately 5% decrease in people declaring that they had never used UGS [70]. In subsequent studies conducted after the second coronavirus wave in Italy (May–June 2021), a 7.7% increase was observed in the frequency of use of UGS sites close to home, and a 16.5% decrease was observed in the percentage of those who reported visiting a UGS once a week [71]. A survey on the adult population in Norway conducted by Litleskare and Calogiuri et al. [28] found that 32% of respondents decide to relax in green spaces more often and 12% less often. Among respondents, women, younger people, and people from higher-income households increased their activity. Using data from Google Mobility Reports in Norway, a 19% increase in park visits was found [66]. However, using the same global data, different behavioral trends were found in different parts of the world [72]. The authors explain this by the influence of political, environmental, social, and personal factors.

The literature contains work that does not confirm the increase in the frequency of visits to UGS areas. One example is the work of Khalilnezhad et al. [55], in which the percentage of respondents who never visited a UGS increased from 4% before the pandemic to 42% during the pandemic. The authors explain this decline through the extent of restrictions introduced and a significant restriction on movement. As an additional explanation, they cite fear and anxiety about getting sick, which was more likely in crowded places such as city parks. According to surveys conducted in Italy and Spain, two countries that were heavily affected by the first wave of coronavirus in the spring of 2020, nearly two-thirds of people who regularly use UGS have abandoned this form of activity. However, as the authors point out [22], the need for contact with nature did not decline, and the restrictions introduced had a decisive influence. Curtis et al. [73] also confirm that during the #stay-at-home rule, park visits decreased by an average of 26% in the U.S. Larson et al. [74] examined reasons for students putting away activities in city parks and pointed out accessibility of parks, risk of illness, and negative thoughts and feelings. A decline in the number of UGS visitors was also observed in downtown parks in Sapporo, Japan, in 2020. In this case, the authors explained the decrease through the recommendation to stay at home and the high level of Japanese culture in terms of legal norms and regulations [25].

An increase in attendance or frequency of visits was also observed in suburban forests during the pandemic. In a study by Bamwesigye et al. [37], conducted in the Czech Republic in a suburban forest complex (Masaryk Forest Křtiny) near Brno, there was an increase of more than 8% in the average daily number of visitors to this area in 2021 compared to the 2015–2018 average. This increase was observed on all days of the week. Additionally, in the Czech Republic, in the suburban forests of Zlín, camera traps recorded a 151% increase in visitors in 2020 (the 1st year of the pandemic) compared to 2019 [62]. However, the sample in this study was small, comprising 127 individuals in 2020 and 84 in 2019. Jarský et al. [43]

showed that the annual number of forest visits in the Czech Republic doubled when comparing 2020 with the 1994–2019 period. The number of people not using forests decreased (decrease from 12.8% in 2019 to 7.8% in 2020), while the percentage of people who reported visiting the forest at least once a month increased from 56.7% (2019) to 70.5% (2020). In one of the first publications on the effects of the pandemic on forest recreation, Derks et al. [21] confirmed the increased interest in forest recreation. Using pyroelectric sensors, they showed that nearly 140% more people visited forests in the suburbs of Bonn, Germany, during the lockdown period than before. Remarkably, the number of visitors decreased again after the restrictions were lifted. Weinbrenner et al. [23] also observed an increase in weekly visits from 2.7 to 4.2 in forests on the outskirts of Freiburg (Germany). This increase mainly affected young people, women, and people from households with more than three people. Grima et al. [75] emphasized that the lockdown period in the U.S. (Burlington, Vermont) brought a nearly 70% increase in visits to green spaces and urban forests. In their study, approximately ¼ of the respondents said they had never or very rarely spent time in green spaces before the pandemic. Using data from 53,000 users of the STRAVA portal, Venter et al. [45] observed a 240% increase in activity during the five weeks of the lockdown (spring 2020). This increase continued until the turnaround in June and July, when activity returned to baseline levels. In the fall of 2020, another 89% increase in activity was recorded. The analysis of the routes registered in the portal showed that users decreased their activity in the built-up part of the city (residential and commercial areas). Moreover, the use of UGS, including forests and legally protected areas, increased. Chen et al. [76], who studied visits to urban and suburban forests in Taiwan, emphasized that indicators related to COVID-19 (the unblocking index and social distancing index) had a significant impact on changes in community mobility. During the lockdown, the number of forest visitors decreased, while during the reduction of restrictions, an increase was observed. Partially different results were obtained by Ciesielski et al. [47] for three forest complexes in Poland. During the first lockdown, an increase in individuals registered by the pyroelectric sensor was observed only in one forest complex in the forests on the outskirts of Gdańsk compared to 2019. In complexes outside major cities, there was a decrease of 47% and 84%, respectively. In their analysis, the authors also examined in detail the changes in society's behavior during the different phases of the pandemic and introduced the restrictions. During the first lockdown in the spring of 2020, the Polish government imposed a temporary ban on entering green spaces and forest areas. During this period, the number of visitors to the forests of Gdańsk decreased by more than 80% and to the other two areas by 50% and 97%, respectively. It is worth noting that in the autumn of 2020, during the second lockdown, when some of the restrictions were of a different nature than in the spring, all complexes recorded an increase in the number of visitors from 14 to 64%, depending on the research object. Grzyb et al. [46], analyzing Instagram data from the period of the first lockdown (spring 2020) in Warsaw, Poland, indicated a decrease in interest in historic city parks and an increase in forests within the city borders and in the suburbs. This finding confirms the important role of forests as recreational sites during a pandemic. The importance of forests during the pandemic was indirectly highlighted by Falk et al. [49]. In their analysis of tourism indicators (arrivals, overnight stays, length of stay) for Bavaria (Germany), they showed that the share of forests in the local areas had a significant influence on the choice of recreation destination in 2020 and 2021. However, the influence was lower in the summer of 2021.

Numerous restrictions also led to a change in daily habits. During the pandemic, some studies observed a change in the timing of activities in forest areas in relation to hours or days of the week. Cukor et al. [62] recorded a 9% increase in the percentage of forest visitors from Monday to Friday (from 67 to 76%). Similar observations during the lockdown were made by Ciesielski et al. [47] and Derks et al. [21]. Derks et al. [21] indicated that prior to the lockdown, the number of visitors before and after work was similar on a daily basis. During the lockdown, there was a visible peak in activity between 5 and 7 p.m. The number of visitors to the forests was almost twice as high as that in the morning. The

change in the number of visits was influenced not only by constraints on daily rhythms, but also by the desire to avoid crowds during current peak hours [77]. The amount of time spent in green spaces has also changed. Nearly two-thirds of respondents in the research of Weinbrenner et al. [23] reported that their visit to forests lasted longer than before the pandemic. Only 4% of respondents indicated the opposite.

**Table 1.** Main list of publications with information about demand for recreation in nature.

| References | Increased | Decreased | Description | Area |
|---|---|---|---|---|
| Bherwani et al. [70] | 9% | | Increase in the number of people who use them daily | UGS |
| Isabella et al. [71] | 7.7% | | Increase in the number of people who use places close to their place of residence | |
| Litleskare and Calogiuri et al. [28] | 32% | 12% | Increase in the percentage of people who use the park more frequently | |
| Venter et al. [66] | 19% | | Increase in the number of visits in the park | |
| Khalilnezhad et al. [55] | | 38% | Increase in the percentage of non-users | |
| Ugolini et al. [22] | | 66% | Decrease in the percentage of people who used the UGS regularly and, now, no longer use it | |
| Curtis et al. [73] | | 26% | Increase in the number of visitors | |
| Grzyb et al. [46] | | | Decrease in interest in areas of historic city parks | |
| Larson et al. [44] | | 56% | Survey respondents indicated that they had stopped or decreased park use | |
| De Meo et al. [69] | 7.7% | 16.5 | Percentage of people who reported visiting the park once a week decreased; percentage of people who use it daily increased | |
| Bamwesigye et al. [37] | 8% | | Increase in the average number of daily visitors | Urban and suburban forests |
| Cukor et al. [62] | 151% | | Increase in the number of visits | |
| Jarský et al. [43] | 5% | | Decrease in the percentage of people who do not use the wooded areas | |
| Derks et al. [21] | 140% | | Increase in the number of visitors | |
| Weinbrenner et al. [23] | 1.5 | | Increase in the frequency of weekly visits | |
| Grima et al. [75] | 69% | | Increase in green spaces and urban forests | |
| Venter et al. [45] | 240% | | Increased activity during lockdown | |
| Ciesielski et al. [47] | | 50%–97% | Decrease during the ban on entry to green spaces and forest areas | |
| Ciesielski et al. [47] | 14%–64% | | Increase during the second lockdown | |
| Grzyb et al. [46] | increase | | Increased interest in urban and suburban forests | |
| Day [38] | | 17% | Decrease during the lockdown | Various |

In summary, among the factors influencing the change in demand for recreation in green spaces and forest areas, restrictions are primarily indicated (Figure 5). According to the researchers, in some way, they forced the search for new places for recreation. The restrictions took various forms, sometimes including a ban on movement or a ban on entering green spaces and forest areas. Thus, the volume of visitation in UGS and forest

areas was controlled by national or regional regulations. Litleskare and Calogiuri [28], regarding the reasons for the increase in the number of visits, mentioned the effects of constraints, such as the following: seeking an alternative way of spending time in relation to the gym and sporting events, more free time, more "flexible days", more free time with family and friends, remote work and learning. Regarding the motives, they mentioned the desire to play sports and stay physically active, to meet family and friends, to reduce stress and dispel worries, to deal with the pandemic reality, as well as the need to find peace and quiet. The reasons mentioned by the authors could also be barriers to accessing green spaces. Introducing very restrictive limitations, with a total ban on leaving the house or a limited distance, undoubtedly led to a decrease in activities. Individuals' internal reasons could also include anxiety and fear of disease, as well as negative emotions, constituting an effective block against the use of green spaces. In addition, decisions about using green areas were influenced by sociodemographic factors such as income, gender, race, and education [45,74]. However, these variables also varied by country of study [28]. From the perspective of the managers of green spaces and forest areas, it is important that society rediscovered the possibility of contact with nature. Society noticed and appreciated the benefits of green spaces, which may lead to specific expectations of society regarding the form of green area management and administration in the future.

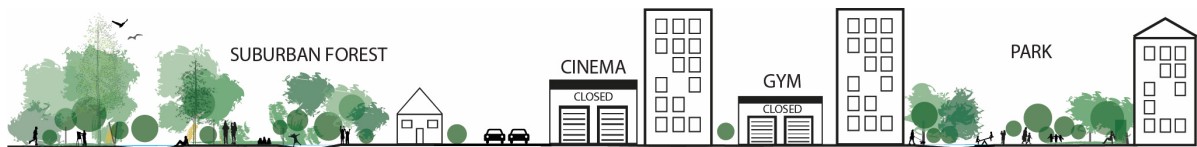

**Figure 5.** Factors and motives determining visits to green spaces.

### 3.2.2. Contact with Nature and Mitigating the Negative Effects of COVID-19

There is no doubt that the COVID-19 pandemic and introducing restrictions had negative effects on people's health and well-being [78]. Loneliness and isolation triggered emotions such as fear, helplessness, anger, insecurity, and concern for health and loved ones [79,80]. Long-term exposure to stressors subsequently leads to insomnia, attention deficit hyperactivity disorder, and depression [81]. This was particularly evident during lockdowns [82]. Contact with nature has become one of the means to mitigate the negative effects of the pandemic related to mental health [83,84].

A confirmation of the significant role of contact with nature is found in the studies on the factors that motivated people to visit green spaces and forests during the COVID-19 pandemic. These include the following: the need to find rest [57,75,85], stress relief [28,86], seeking fresh air and willingness to get some fresh air [68], and health benefits [87,88]. Weinbrenner et al. [23] noted that visitors to forest areas did so consciously or unconsciously primarily for health reasons. It should be worth remembering that green spaces enabled the maintenance of social contacts while maintaining social distance due to the constraints [75]. In this way, they were able to reduce the negative effects of restrictions. Additionally, the activities in the green spaces show that they have become a new place for people to take care of their health. Litleskare and Calogiuri [28] indicated that 39% of respondents increased their activities in green spaces related to walking. Isabella et al. [71] observed an increase in physical activity in urban green spaces in their research on residents of Italy. In the Czech Republic, the percentage of people visiting forests several times a month for walking, contact with nature, and sports increased in 2020 compared to 2019 [43]. Huang et al. [58], in an analysis using data from Twitter (now X), indicated that there was a significant increase in active recreation by running or walking in parks in New York. In Central Park, the increase was nearly 200%. A UGS study in India found that 85% of respondents increased their yoga-related activities after the lockdown [70].

Studies on the effects of greenery on health show that even a green view of the window and access to a private garden at home reduce symptoms of depression, stress, and loneliness [89]. Moreover, life satisfaction increases. In their study, Wajchman-Świtalska et al. [80] examined the relationships between three strategies (acceptance strategy, active coping strategy, and avoidance strategy) for dealing with problems and the frequency of forest visits. The authors indicated that individuals who spent more time in forest areas chose the acceptance strategy more often than individuals who visited the forest only on weekends. In addition, the authors demonstrated that the remaining two strategies were predictors of increased forest use during the pandemic. Lee et al. [27] confirmed in their study that a greater frequency of visiting forest areas leads to greater stress resistance. The authors argued that this was due to the phytoncide released by the trees. It should be emphasized that, according to the authors, the reduction in stress levels leads to an increase in the motivation to recreate in forest areas. In a study conducted in Tokyo, Soga et al. [89] found that stress levels and depression were negatively correlated with the frequency of visits to green spaces. Lee et al. [86], while examining the role of three different forest areas in relation to forest management (urban forest, recreational forest, national park), found that the type of forest did not affect the ability to relieve stress. Puhakka [52] pointed out the positive aspects of contact with nature, such as: calming, relaxation, improved concentration, increased energy, or visible improvement in health (reduction in headaches, more sleep, lower pulse rate). It is important to note that respondents in this study indicated that they were influenced not only by visual elements, but also by sound, smells, and touch. Korpilo et al. [42] divided the respondents into two groups. The first group used similar places for recreation before and during the pandemic. This group showed a decline in self-rated health. The second group, which mapped significantly more places for recreation, was characterized by a higher level of self-assessment of health. This group consisted mainly of older people who were proportionately more likely to work, have a partner, and have children than representatives of the first group. Several researchers, such as Chen et al. [76], have pointed out the need for further, detailed research on the relationship between forest ecosystem services and human well-being. They emphasize that this is due to the fact that society appreciates the health-promoting role of forests during a pandemic.

To sum up, the vast majority of studies indicate the positive effects of green areas on human health and well-being during a pandemic. The key element for society to derive health benefits from contact with nature is its availability. Availability is understood at different scales and distances. We can talk about green spaces seen from the window, green spaces in a private garden [89], green spaces growing along the way to work or school [90], and generally available green spaces in a certain area and at a certain distance from the place of residence [91,92]. Therefore, ensuring an adequate availability of green area is a challenge, especially for local governments in cities where it is typically limited. The health benefits of contact with greenery have also been recognized in EU documents, such as the European Biodiversity Strategy to 2030, entitled "Bringing nature back into our lives", and the EU Forest Strategy (2030) [93].

### 3.3. Cities' Spatial Policy–Post-COVID-19 Challenges and Lessons Learned

The COVID-19 pandemic has further highlighted the need for sustainable urban spatial management [57]. The key aspects seem to be the following: the availability of green spaces in different areas of cities and for different social groups [22]; public participation in the spatial planning process, e.g., in the development of recreational infrastructure and the design of green spaces (e.g., preservation of existing greenery, design of street greenery, introduction of new forms of greenery, including vertical greenery) [57,94]; management of forests around cities [85].

The issue of green area availability is not a new topic. So far, many indicators of green area availability (distance and travel time) have been developed [95] as well as concepts, i.e., the 15 min city [96] and 3-30-300 [97,98]. None of these indicators are universally applicable, and different countries may use different indicators. The pandemic also highlighted the

problem of equal access to green areas for different social groups [56]. Ugolini et al. [22] pointed out that the need for contact with nature is so great that residents of areas without UGS areas at their residence decide to travel significant distances outside or inside the city to use green spaces.

On the other hand, limited access to green area in the place of residence means that older, less wealthy people, and people from ethnic minorities more often decide to stay at home [99–101]. Therefore, special attention should be given to the design of green spaces when planning urban development, taking into account the needs of different user groups. In urban areas, it is important to ensure easy access to green spaces, not only in terms of distance from home, but also in terms of well-organized public transport. As Puhakka [52] points out, this is especially important for vulnerable groups such as young people, for whom access to urban and suburban green spaces should be facilitated. Involving society in the decision-making process is an important element that can support the process of urban regeneration in the context of climate change adaptation and societal expectations for urban green area. Through public participation, it is possible not only to gain knowledge of residents' needs, but also to foster their sense of connection to the environment and acceptance of necessary actions in the future [57,102,103]. It should be remembered that the decision-making process should be conducted with the consideration of as many stakeholders as possible [94]. Only by building broad understanding among groups can the risk of future conflict be minimized [45]. Conscious decision-making about green design in the city also requires data that can support this process. These data should cover aspects such as ecosystem services (potential and supply), the condition of green spaces, and benefits to society, including health benefits, among others [45,70,104,105]. The availability of reliable data is especially important in the post-pandemic period, as the habits and needs of different user groups may have changed [25–42]. Yamazaki et al. [106] emphasized that it is essential for spatial planners to consider short- and long-term changes in user needs. Khalilnezhad et al. [55] believe that not only the quantity but also the quality of available green areas is important. Yin et al. [105] presented practical mechanisms for establishing and managing urban forests to maximize the benefits of their health-promoting functions. They divided their considerations into three groups: physiological, psychological, and immunological.

Adapting green spaces in cities to the challenges of the pandemic means not only planning new spaces, but also protecting existing ones. Cities should raise funds to preserve, restore, and protect existing green resources from excessive building and infrastructure expansions [94]. Uncontrolled processes associated with urban development are one of the greatest threats to green spaces, especially in large metropolitan areas that attract new residents owing to their advantages [64].

Future city greening activities to improve the lives of residents and provide resting places during possible future pandemics are consistent with documents developed in recent years at the international level [22,107]. The United Nations Sustainable Development Goals No. 3 ("Good Health and Well-Being") and 11 ("Sustainable Cities and Communities") indicate that an appropriate green area design is necessary to ensure sustainable development and urban resilience [108]. At the European Union level, the initiated recovery plan also recognizes the need for urban reconstruction, green transformation, and strengthening the role of green and blue infrastructures [109]. Venter et al. [45] add that understanding the role of green spaces in recreation is critical to achieving these goals.

To sum up, cities will have to undergo a lengthy and costly transformation in order to meet society's requirements in the area of urban greenery. It is, therefore, necessary to provide cities with the legal and financial means to accomplish this change. At the same time, cities need objective and reliable data on the basis of which they can make appropriate decisions. The process of change may also require the support of the public, which is why the appropriate involvement of society in the decision-making process is also important.

### 3.4. Forests and Protected Areas–Adaptation to Social Needs

Society's turn toward nature [23] has led to new challenges for forest managers, or existing challenges have become more important. The first is to balance the various functions of the forest with society's expectations. During the pandemic, forest areas were visited by new user groups, including the youth, whose perceptions of forests and forest management may differ from those of regular visitors [85,110]. The apparent decline in the acceptance of timber harvesting combined with the increased number of forest visitors may lead to conflicts due to different expectations of forest management [47]. Therefore, foresters should be trained not only in appropriate forest management planning for timber harvest, but also in optimizing other forest functions, including forest recreation opportunities [85]. A new perspective on forest management is especially important in peri-urban areas, where public pressure on forests is high and the pandemic has increased the need for contact with nature [23,69]. Koprowicz et al. [111] emphasize that suburban forests should provide many ecosystem services that are important from a society perspective. The authors also noted the need for research on socially acceptable forms of forest management (e.g., thinning instead of logging) and the need for recreational infrastructure. Similarly, Weinbrenner et al. [23] added that the timing of logging can be adjusted to the visitation level in the area. Guidelines for managing suburban forests also recommend leaving tree stands adjacent to heavily traveled roads to avoid disrupting the visual experience provided by timber harvesting [112]. Pichlerova et al. [65] point out that dialogue between different stakeholders (forest owners, managers, local authorities, and society) should be a tool to develop more nature-oriented forest management practices. The application of such principles could support the use of forests as a tool to mitigate the negative impacts of the pandemic on health and well-being [36].

One of the main reasons for choosing to visit the forest both before and during the pandemic was the opportunity to find peace and admire the beauty of nature [85]. An increase in the number of visitors may make it more difficult to implement these conditions. Therefore, it can be challenging for forest managers to plan recreational infrastructure to distribute visitor flows to different parts of the forest. This is necessary to minimize the risk of conflicts between different user groups [113]. Another aspect was highlighted by Dudáková et al. [61]. According to them, increased recreational interest and a larger number of users also mean the need for investments to maintain the proper condition of trails.

According to Ferguson et al. [114], management agencies should consider the following factors when planning recreational infrastructure: congestion, conflicts, site accessibility, and equal treatment of all visitors. Ferguson et al. [115] also indicated that managers of protected areas may adopt policies that assume some visitation limits which are acceptable from social and environmental perspectives. These policies may include direct and indirect actions that affect visitation (e.g., introducing entrance fees). At the same time, they emphasized that the choice of measures should be preceded by a thorough analysis of the impact not only on visitors and the environment, but also on the residents of the surrounding areas. Grima et al. [75] emphasized that management strategies should be tailored to the targeted area and its characteristics.

Ferguson et al. [114] argue that because of increased recreational use in natural areas, it makes sense to educate the public on caring for the natural environment and minimizing potential damage and risk from recreation. The potential impacts of excessive pressure on ecosystems were also mentioned by Ferguson et al. [115], who identified visitor education as one of the appropriate indirect management strategies (e.g., "leave no trace"). Baranowska et al. [68] also pointed out the need for education, emphasizing that tourists are not always aware that their activities may pollute a particular area. According to them, tourists attach more importance to the benefits they derive from their trips than to the potential impacts of their activities. The issue of excessive visitation levels and methods of eliminating negative impacts in environmentally valuable areas was also raised by McGinlay et al. [116], who pointed out ways to mitigate hazards during a pandemic. They

focused on online education campaigns to raise awareness among visitors and promote environmentally friendly behaviors. To summarize, forest managers must take into account the societal expectations of forests that have arisen during the pandemic. However, this will not be an easy task considering the current challenges for forestry due to climate change or the ongoing legislative changes at the EU level. To reconcile the expectations of different stakeholders, decisions need to be made at different spatial levels, from a single forest complex to the forest district and national levels. As with UGS, decision-makers need to be provided with appropriate data and methods to develop the best solutions for the management of forest areas [117].

## 4. Conclusions

The importance of green spaces as recreational areas for society and the health-promoting effects of greenery on human health and well-being have been known for many decades. During the pandemic period, this importance was reinforced, and society refocused its attention on the value of green spaces. Despite the high number of works presented in this review, many questions remain unanswered and require further research. The first concerns the "permanence" of the changes. Will the increased interest in green-area recreation continue after the pandemic? Will the increase in the number of visitors during the pandemic lead to long-term changes in human–nature relationships? Another question concerns the impact of green spaces on minimizing the negative, long-term health effects of COVID-19 [28], potential social conflicts, public participation, or the long-term monitoring of the recreational use of green spaces and forests. From a decision-maker's perspective, it is extremely important for science to provide reliable data for strategic planning and to develop quantitative and qualitative methods with which such data can be obtained.

**Author Contributions:** Conceptualization, M.C.; methodology, M.C.; formal analysis, M.C.; investigation, M.C. and K.T.; writing—review and editing, M.C., P.G., K.T. and F.S.; visualization, F.S. and M.C.; project administration, M.C. and K.T.; funding acquisition, M.C. and K.T. All authors have read and agreed to the published version of the manuscript.

**Funding:** Funded by the National Science Centre, Poland, under the OPUS call in the Weave programme (project No. 2021/43/I/HS4/01451) and the Austrian Science Fund FWF (grant No. I 6083-G).

**Conflicts of Interest:** The authors declare no conflicts of interest.

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
