# Peer review of "Unveiling the Essential Role of Green Spaces during the COVID-19 Pandemic and Beyond"

_forests, doi:10.3390/f15020354_

Round 1

Reviewer 1 Report

Comments and Suggestions for Authors

This paper presents a systematic review of research published on the role of green spaces within the context of the COVID-19 pandemic. This is a timely review of published literature since the start of the pandemic and as such presents a relevant contribution to the knowledge on links between green spaces and health and wellbeing. The review set out to address four main questions around the research methods used to determine social importance of green spaces, changing recreational use, impacts of use of green spaces on health and wellbeing, and finally the changes caused to planning and decision making based on the changed engagement with green spaces. While there are some limitations in the methods described, e.g. additional search terms and combinations could have been included which would have made this a more comprehensive review, the overall approach seems appropriate and in line to address a knowledge gap in relation to use of green spaces during Covid. There is currently no discussion section, but the results section contains elements of discussion that could be reconsidered to serve as such. The information and discussion presented are relevant and address the questions asked, but should be presented more appropriately.

Specific comments 

Abstract:

While the abstract sets out the premise and methods of the article very well, results, conclusions and lessons learned are not included. It would be great so see the main findings included in the abstract. 

Introduction:

Overall, the introduction sets the scene well in relation to impacts of Covid-19, but it could introduce the recreational use aspect further. It would be useful to provide a definition of green space and introduce ecosystem services as this is one of the search terms for the review.

Line 47-48: and some, one might say, thanks to the pandemic, have decided to make such visits for the first time. [recommend refraining from inclusion of personal judgement]

Lines 55 – 57: could examples of such challenges be included?

Materials and Methods:

The methods generally follow standard approaches for strategic reviews and are as such fine, although search terms could have been better defined and the inclusion of forests as specific search term but no other variants of green spaces such as parks is not fully clear as this would have possibly given a more comprehensive picture.

Lines 74 – 76: More clarity on search terms could be provided, e.g. was greenspace included as a variation of green space, were green space and green area considered loose phrases and what about * wildcards?

Additional information on inclusion/exclusion criteria should be provided, in particular reference to language would be helpful. It is not clear that literature in languages other than English were included in the search until you note that all literature except that in English and Polish was excluded.

Lines 84 – 92: confusing phrasing, this currently reads like all papers relating to the topics in bullet point list were rejected, surely that was not the case?

Figure 1: Check keywords, the figure includes green area, which is not listed in lines 74 – 77 as a search term. Check the number of publications identified through your systematic search again, figure states n=363, but text above (lines 81 – 82) list 163 and 180, which equals n=343.

Figure 1: consider referring to initial stage as ‘systematic literature search’ rather than ‘systematic literature review’ as you have not yet reviewed literature at this stage.

Line 93: In total, there were 62 items of literature that did not cover the topic relevant topics. [I would argue that there is more than one topic here and hence would refer to topics]

Results:

The overall presentation of results is appropriate and the visualization of results works. However, this section contains elements of discussion and then there is no discussion section as such. For this to be a results section, the recommended course of action is to clearly separate results and discussion elements.

Line 103: the phrase ‘practically every continent’ should be replaced with specific detail as the subject has been the topic of research on five out of seven continents. No publications were found for South America and Antarctica.

Lines 113 – 123: detail on how many papers covered the listed topics or relevant green space types should be included here. Also, the differentiation between urban/suburban forests and other urban green spaces is not clear.

Line 116: Abbreviation UGS was not previously introduced – first mention of urban green spaces is in line 46 and (UGS) should be included there.

Line 120: the different phases of the pandemic are not clear, what is meant by this?

Line 135: Refer to Survey Research as the title rather than Survey, which can have broader interpretation.

Line 137: Not sure [40] this is the most appropriate reference here.

Line 137/138: replace commonly with frequently, which is more appropriate here as this is based on their counted observations.

Lines 141 – 143: cannot understand what is meant by this. May require more discussion.

Lines 143 – 145: the different times should be outlined, how many surveys were carried out during lockdowns, post lockdown etc, whatever relevant periods there have been identified.

Lines 145 – 206: Again, how many articles covered the different topics? These section provide overviews of the different methods used, but do not present detailed results which should be the starting point for each section.

Table 1 should be included in the Results section and similar tables could be created for the results in sections 3.3 and 3.4

Discussion:

There is currently no discussion section as such, but sections 3.2 – 3.4 could be reconsidered as such, if preceded by a short discussion of results presented in 3.1, which is essentially the conclusion of section 3.1, lines 207 – 219.

Comments on the Quality of English Language

Overall, the language is good, some minor corrections are required in places to ensure correct grammar is used.

Author Response

Dear Reviewer,

Kindest regards,

Authors

Reviewer 2 Report

Comments and Suggestions for Authors

The paper is a typical secondary research document aiming to run a systematic literature research on the social needs for urban green areas and forests with special concern for the 2020-2023 pandemic. The method is relevant for such a publications review. The selection of 161 papers focusing on green areas recreation use during COVID-19 pandemic underlines the importance of green infrastructure's social and health context. Then, the second-stage analyses resulted in 66 papers, a relevant sum for the detailed full-text analyses.

However, the introduction failed to give an overview of the context of social and human well-being, and the regular use of green areas. The list of references contains 106 documents published directly in the pandemic period, except for 3 items which belong to the analyses method part. Many analysed papers contain comparative analysis data of the earlier and the pandemic period. Still, the introduction should be based on a summary of the relevant literature on public open space recreation and its social and health contexts. From among the long list of authors and papers, I would call the authors' attention to the complex, multi-scientific papers of C.W. Thompson at al. 

The Result part is well-structured and informative in many aspects. It starts with the methodological analyses of the original research papers which is a useful help for further site analyses and research. The main findings of the relevant papers obviously conclude the increased social demand for recreation in green spaces. Green area availability within urban landscape architecture is one of the traditional and the most often analysed aspects which could or even should be taken into account in urban development programs and green infrastructure supply projects. On the other hand, as stated in some original papers, the increased interest in green areas among urban residents may cause overuse and other ecological problems.

These important findings' summaries lay stress on the time scale of the given secondary research paper. All sorts of green infrastructure research papers could strengthen the scientific fundaments of urban landscape architecture and the overall urban policy. All research data, as convincing as it is possible might help to make a step toward a more sustainable urban feature. While the authors comment in the conclusion that the increased social need for urban or natural green as an effect of the COVID pandemic has been proven, a further research question could be the future stability of this green interest. Answers and convincing results are awaited to create a scientific statement for urban planning and sustainability development aims. The present review paper offers the main aspects and methods of further research. 

Comments on the Quality of English Language

Not being a native English speaker, the paper is well readable but moderate editing is needed

Author Response

Thank you for reviewing our article and providing a positive rating. In the review written by the reviewer, we found one point that we have included in the text of the article.

Comment 1:
“However, the introduction failed to give an overview of the context of social and human well-being, and the regular use of green areas. The list of references contains 106 documents published directly in the pandemic period, except for 3 items which belong to the analyses method part. Many analysed papers contain comparative analysis data of the earlier and the pandemic period. Still, the introduction should be based on a summary of the relevant literature on public open space recreation and its social and health contexts. From among the long list of authors and papers, I would call the authors' attention to the complex, multi-scientific papers of C.W. Thompson at al.

Response:

Thank you this suggestion. We added one paragraph: Forests are an integral part of the landscape and have fulfilled numerous functions for centuries. Originally, the production function dominated, but since the beginning of the 20th century the non-productive functions of forests (social and protective) have also been increasingly recognized [1]. Urban green spaces, which include all managed green spaces (parks, squares, green areas) and unmanaged green spaces (urban forests, wood-lands, wastelands), are also an important for society and important element of the urban composition [2-3]. The role of forests and green spaces for society is reflected in the concept of ecosystem services. According to the international classification of ecosystem services, forest areas are considered capable of providing over 100 ecosystem services [4]. Ecosys-tem services are divided into four groups: cultural, regulating, provisioning and support-ing [5]. The need for services and the demand on them are diverse [6]. In the study by Hegetchweiler et al [7], it was shown that non-material ecosystem services are more im-portant to society, especially for residents of urban areas, than those that provide specific products. Among the cultural services, those associated with the possibility of active and passive recreation in green spaces and contact with nature are of particular importance [8]. As research published over several decades has shown, contact with nature has a pos-itive effect on human health and well-being [9-10]. Human health is also influenced by regulating ecosystem services that reduce the risk of lung diseases through air purifica-tion, among other things [7]. However, in urbanized areas, access to green spaces is usu-ally limited or unevenly distributed across urban areas. Nevertheless, publicly accessible green spaces and forests are important public spaces that serve as places for daily recrea-tion [11]. The importance of these places was demonstrated in the early days of the COVID-19 pandemic.